# A One Health Comparative Study of MDR *Escherichia coli* Isolated from Clinical Patients and Farm Animals in Satu Mare, Romania

**DOI:** 10.3390/antibiotics14111157

**Published:** 2025-11-14

**Authors:** Iulia-Maria Bucur, Anca Rus, Kalman Imre, Andreea Tirziu, Ionica Iancu, Andrei Alexandru Ivan, Alex Cristian Moza, Sebastian Alexandru Popa, Ionela Hotea, Emil Tirziu

**Affiliations:** 1Department of Animal Production and Veterinary Public Health, Faculty of Veterinary Medicine, University of Life Sciences “King Mihai I”, Calea Aradului 119, 300645 Timisoara, Romania; mekker.anca-sm@ansvsa.ro (A.R.); kalmanimre@usvt.ro (K.I.); andreialexandru.ivan.fmv@usvt.ro (A.A.I.); alex.moza@usvt.ro (A.C.M.); sebastian.popa@usvt.ro (S.A.P.); ionelahotea@usvt.ro (I.H.); emiltirziu@usvt.ro (E.T.); 2Ophtalmology Department, Faculty of Medicine, “Victor Babes” University of Medicine and Pharmacy, Piata Eftimie Murgu 2, 300041 Timisoara, Romania; andreea.tirziu@umft.ro; 3Department of Infectious Diseases and Preventive Medicine, Faculty of Veterinary Medicine, University of Life Science “King Mihai I”, 300645 Timisoara, Romania; ionica.iancu@usvt.ro

**Keywords:** multidrug resistance, *E. coli*, human, animals, One Health

## Abstract

**Background/Objectives**: Multidrug-resistant (MDR) *Escherichia coli* is a critical One Health challenge, with rising resistance in both humans and animals. The present study aimed to compare antimicrobial resistance (AMR) profiles of *E. coli* isolates from hospitalized patients and food-producing animals in Satu Mare, a county located in northwestern Romania. **Methods**: Between 2022–2023, 701 samples were collected, leading to 571 non-duplicate *E. coli* isolates (420 human, 151 animal). Human strains were recovered from 21 hospital departments and originated from feces, urine, blood, sputum, ear secretions, cerebrospinal fluid, purulent wound secretions, and puncture fluids. Animal isolates were obtained from ceca collected at local slaughterhouses serving farms in north-west Romania, including samples from turkeys, broilers, and pigs. Antimicrobial susceptibility testing was performed against eight antimicrobials (amikacin, ampicillin, cefotaxime, ceftazidime, cefepime, ciprofloxacin, gentamicin, sulfamethoxazole/trimethoprim) using standardized methods. Resistance classification followed international definitions of MDR. Statistical associations between host species and resistance were assessed with chi-square tests. **Results**: Resistance levels were consistently higher in *E. coli* strains isolated from animals compared with those from humans (*p* < 0.05). Among human isolates, resistance to ampicillin (41.9%), ciprofloxacin (41.4%), and sulfamethoxazole/trimethoprim (45.7%) approached, but did not exceed 50%. In contrast, *E. coli* strains recovered from animals showed markedly higher resistance, exceeding 50% for ampicillin (78.8%), ciprofloxacin (65.6%), and cefotaxime (55.0%). Amikacin retained full activity against all animal isolates, whereas 2.8% of human strains were resistant. Overall, multidrug resistance (MDR) was observed in 70.0% of *E. coli* isolates from humans and 79.7% from animals, with the highest resistance burden in pig-derived isolates. **Conclusions**: The study underscores the veterinary sector as a key contributor to the maintenance and spread of MDR *E. coli*. Even in clinically healthy animals, resistance levels exceeded those observed in human isolates. These findings emphasize the need for coordinated One Health monitoring and stricter antimicrobial use policies in livestock to reduce transmission risks across human and animal populations.

## 1. Introduction

The emergence of multidrug-resistant (MDR) pathogens represents a tangible and continuously expanding threat, with major implications for both human and animal health. This phenomenon is primarily driven by increasingly frequent therapeutic failures and the high costs associated with treatments, as well as by significant losses in livestock production. The risks are further amplified by the widespread use of the same classes of antimicrobial agents in both human and veterinary medicine, thereby facilitating the emergence and selection of resistant strains [1,2,3].

Most research efforts focus on global trends in antibiotic use, as well as on investigating the degree of bacterial resistance to antimicrobials and the associations between antibiotic use and resistance rates [4,5].

The irrational consumption of antibiotics represents the main selective pressure promoting the development of resistance, both in human and veterinary medicine. Currently, numerous studies demonstrate the existence of a clear correlation between the often chaotic application of antimicrobial agents and the spread of antimicrobial resistance, especially among *E. coli* strains [6,7,8].

When it comes to *E. coli*, most studies on the prevalence and severity of antimicrobial resistance are limited to specific countries or regions and tend to provide only a snapshot of the relationship between antimicrobial use and the emergence of resistance. Collectively, however, these findings represent only the “tip of the iceberg” of the broader antimicrobial resistance phenomenon [9,10,11].

Moreover, in case of *E. coli* strains, a critical link between human and animal populations in the development of antimicrobial resistance is the extensive use of antimicrobials, both for therapeutic and prophylactic purposes, as well as their environmental persistence [12,13,14]. The high concentrations of antibiotics presently identified in the environment, recognized as a serious form of pollution, are mainly the result of uncontrolled use of these substances in both livestock sector and human medicine [15,16].

Antimicrobial resistance is broadly acknowledged as a global One Health challenge, driven by interrelated aspects of human, animal, and environmental health. Worldwide organizations like the WHO, WOAH or FAO have highlighted the need for coordinated antimicrobial resistance surveillance and control strategies covering all sectors [17,18].

In Romania, antimicrobial consumption remains high in both human and veterinary medicine. In human healthcare, β-lactams (particularly penicillins and third-generation cephalosporins), fluoroquinolones, and macrolides are among the most frequently prescribed classes [19]. In the veterinary sector, tetracyclines, penicillins, sulfonamides, and fluoroquinolones are widely used for therapeutic and metaphylactic purposes, especially in intensive livestock production systems [20]. Such patterns reflect similar selective pressures across sectors and highlight the importance of integrated antimicrobial stewardship within a One Health framework.

The epidemiology of antimicrobial-resistant microorganisms involves a complex network of transmission pathways, both for resistant bacteria and for antimicrobial resistance genes. They circulate between the three major sectors—human, animal, and environmental—under the constant influence of the selective pressure exerted by antimicrobial use. Interactions between these domains contribute to the persistence and spread of resistance, escalating the phenomenon of antimicrobial resistance into a critical challenge for both public and veterinary health [21,22].

From a One Health perspective, the link between antimicrobial use and the emergence of resistance in an increasing number of microorganisms, involved in pathologies both in animals as well as in humans, is becoming more and more evident. This relationship ultimately leads to the more frequent occurrence of multidrug-resistant bacterial strains within populations. However, it remains to be clarified to what extent the human and veterinary sectors are interconnected, such that antimicrobial use in one of them can drive rising resistance levels in the other [23,24,25,26].

According to literature data, the transfer of resistance genes from animal-associated bacteria to those affecting humans follows complex pathways and cannot be controlled solely by reducing antimicrobial use in the veterinary sector. Therefore, an integrated approach is essential to limit the spread of antimicrobial resistance within the human population. Improved hygiene and biosecurity measures along the food production chain, the implementation of alternatives to antimicrobials such as vaccination and probiotics, and systematic surveillance of resistance determinants in humans, animals, and the environment are some of the alternatives that could reduce the spread of the antimicrobial resistance [4,8]. Such strategies align with our study’s objective of identifying the extent of *E. coli* resistance across human and animal sources, thereby emphasizing the need for coordinated One Health interventions.

In Romania, research on antimicrobial resistance in *E. coli* has been largely fragmented and confined to isolated investigations in specific sectors, such as poultry meat, pig and cattle farms, or wastewater. While these studies highlight concerning resistance levels, they do not provide an integrated view across the human, veterinary, and environmental domains [2,27,28,29]. Direct comparative studies that simultaneously address clinical isolates from humans and food-producing animals are absent or without a broader One Health perspective.

This scarcity of comprehensive, cross-sectoral data represents a significant gap in the national AMR landscape. Therefore, this study could represent one of the first systematic attempts to bridge this divide by directly comparing *E. coli* isolates from hospitalized patients and food-producing animals using standardized antimicrobial susceptibility testing and MDR/XDR (extensively drug-resistant) classification. In doing so, it provides a foundation for future One Health research in Romania and opens the way toward a more integrated understanding of antimicrobial resistance at the human–animal interface.

In this context, this research aimed to phenotypically compare the antimicrobial resistance patterns of *E. coli* strains isolated from human patients and food-producing animals collected in north-western Romania during 2022–2023.

## 2. Results

### 2.1. Isolate Distribution by Source, Age, Gender (Human Strains) and Species (Animal Strains)

A total of 571 non-duplicate *E. coli* isolates were recovered during 2022–2023, comprising 420 human and 151 animal isolates (Table 1). From 550 clinical samples, 420 *E. coli* were identified (76.4% culture positivity). The departments with the highest numbers of *E. coli* isolates were Internal Medicine (64/72 samples), Neurology (46/53), Nephrology (48/57), Urology (44/69) and Surgery (34/41), whereas Hematology (2/3), Hemodialysis (2/5), Neonatology (2/4) and Rheumatology (2/5) yielded the fewest isolates.

Age distribution showed the highest proportion of human isolates in patients >65 years (272/420; 64.8%), followed by 41–65 years (108/420; 25.7%), with fewer isolates from 18–40 years (16/420; 3.8%) and pediatric groups (Table 2). By gender, 268/420 (63.8%) isolates were from females and 152/420 (36.2%) from males (Table 2).

Among animals, isolates were recovered from broilers (*n* = 27), turkeys (*n* = 31) and pigs (*n* = 93), totaling 151 isolates (Table 1).

### 2.2. Resistance Profiles of Human and Animal Isolates

In human isolates (*n* = 420), resistance prevalence was SXT 45.7%, CAZ 43.3%, CIP 41.4%, AMP 41.9%, CTX 25.2%, GEN 17.1%, FEP 9.5% and AMK 6.0% (Table 3). All were below the 50% threshold.

The heatmap reveals marked differences in resistance profiles between departments. Nephrology, Urology, and Internal Medicine showed the highest prevalence of resistance to β-lactams (ampicillin, cefotaxime, ceftazidime, cefepime) and fluoroquinolones (ciprofloxacin). Neurology also displayed elevated ampicillin resistance (>65%). In contrast, departments with fewer samples, such as Hematology and Hemodialysis, generally exhibited lower resistance percentages. Notably, resistance to amikacin remained low across all departments, confirming its preserved activity in the hospital setting. (Figure 1).

In animal isolates (*n* = 151), resistance prevalences were AMP 78.8%, CIP 65.6%, SXT 60.9%, CAZ 55.6%, CTX 55.0%, FEP 51.0%, and GEN 25.8% (Table 4). Notably, no animal isolates were resistant to amikacin.

The chart illustrates that pigs consistently exhibited higher resistance levels than poultry, particularly for ampicillin, cefotaxime, and cefepime, where resistance exceeded 50% in pig isolates. Ciprofloxacin and sulfamethoxazole/trimethoprim also showed elevated resistance in pigs compared to broilers and turkeys. In contrast, resistance among poultry isolates was generally lower, with broilers displaying the lowest prevalence across most antibiotics. These findings confirm statistically significant differences (χ^2^, *p* < 0.001) between pigs and poultry for key β-lactams and fluoroquinolones, highlighting the stronger selective pressure associated with antimicrobial use in swine production systems (Figure 2).

The analysis in Figure 3 shows that pig isolates were more likely to be resistant than poultry isolates for most antibiotics, although confidence intervals often crossed 1 (not statistically significant). The strongest association was observed for sulfamethoxazole/trimethoprim (OR = 2.36, 95% CI: 1.20–4.65, *p* = 0.012), indicating that pig isolates had more than twice the odds of resistance compared to poultry. For ampicillin and cephalosporins (cefotaxime, ceftazidime, cefepime), OR values were >1 but not statistically significant, suggesting a consistent trend toward higher resistance in pigs. Ciprofloxacin showed no difference (OR ≈ 1).

### 2.3. Comparative Analysis

Statistical comparisons using the chi-square test revealed a strong association between the species (humans, birds, and pigs) and the antimicrobial susceptibility profiles of the studied strains for the six antibiotics. Therefore, for ampicillin, cefotaxime, cefepime, ciprofloaxicin, sulfamethoxazole/trimethoprim and gentamicin the strains isolated from pigs showed a much higher resistance rate compared to the resistance rate of strains isolated from other species (AMP *X*^2^ (2, *N* = 571) = 60.5, CTX *X*^2^ (2, *N* = 571) = 45.3, CIP *X*^2^ (2, *N* = 571) = 24., SXT *X*^2^ (2, *N* = 571) = 15.6, FEP *X*^2^ (2, *N* = 571) = 117.6 *p* < 0.001 and GEN *X*^2^ (2, *N* = 571) = 6.0, *p* < 0.05). Instead, for ceftazidime (CAZ), the strains exhibited a similar behavior, regardless of the species (*X*^2^, *N* = 571) = 3.4, *p* > 0.05.

Comparing the prevalence of antimicrobial resistance in the two types of populations studied (humans and animals), the data obtained indicate a significantly higher level of resistance in strains isolated from animals (*p* < 0.05). The only notable exception was for AMK, where no resistant strains in animals were found.

The heatmap (Figure 4) highlights that animal isolates consistently display higher resistance levels than human isolates, particularly for ampicillin, cefotaxime, cefepime, ciprofloxacin, and sulfamethoxazole/trimethoprim, where prevalence exceeds 50% in animals. In contrast, resistance in human isolates remains below the 50% threshold for all tested antibiotics, suggesting a relatively more favorable susceptibility profile. Amikacin stands out as the only antibiotic with negligible resistance in both groups, indicating preserved efficacy. This visual comparison underscores the stronger selective pressure exerted in the veterinary sector, even among clinically healthy animals, compared to hospitalized human patients.

The dendrogram shows two main clusters. The first cluster groups β-lactams (ampicillin, cefotaxime, ceftazidime, cefepime), reflecting their shared mode of action and consistent co-resistance patterns. Within this group, cefotaxime and cefepime cluster closely, suggesting parallel selection pressures in both human and animal populations. The second cluster includes ciprofloxacin and sulfamethoxazole/trimethoprim, which often co-occur in multidrug resistance phenotypes due to their frequent combined use in therapy. Gentamicin and amikacin form a separate branch, highlighting their distinct aminoglycoside resistance profile, with amikacin remaining highly effective in animals. Overall, the clustering pattern supports the hypothesis that antibiotic use in veterinary practice exerts selective pressure leading to resistance patterns that mirror those observed in clinical human isolates, particularly for β-lactams and fluoroquinolone/folate inhibitors (Figure 5).

### 2.4. Classification into MDR/XDR Categories

Analyzing the antimicrobial resistance profile and classifying the *E. coli* bacterial strains into resistance categories, based on susceptibility or non-susceptibility to the eight studied antimicrobial substances, was found that of the 420 strains isolated from human patients, 294 strains (70%) were MDR (multi-drug resistance), and 126 strains (30%) were non-MDR (non-susceptible to <3 of the antibiotics). It should be noted that no strain of *E. coli* isolated from patients was 100% susceptible to all antimicrobial substances taken under study.

On the other hand, in the strains isolated from the animal population, no case of amikacin resistance was recorded, unlike *E. coli* strains isolated from the human population, in which 2.8% of the strains were resistant to this antibiotic.

Regarding the strains isolated from animals, these were also grouped into two susceptibility categories, namely 120 strains were included in the MDR category (79.47%) and only 17 strains (11.25%) were classified as non-MDR, with the mention that a number of 14 of the tested strains (9.2%) were 100% susceptible to all antimicrobials studied.

The results obtained from antimicrobial susceptibility testing (AST) revealed important differences between *E. coli* strains isolated from the two investigated populations, respectively, humans and animals. Thus, the comparative analysis shows that the overall prevalence of antimicrobial resistance was higher in the case of strains isolated from the animal population (79.7%), compared to strains isolated from human samples (70%).

This difference is validated by the prevalence values recorded for each antimicrobial substance tested. For example: ampicillin (AMP): 78.8% of strains isolated from animals were resistant, compared to only 41.90% of human strains; ciprofloxacin (CIP): 65.5% resistance in animal strains, compared to 41.4% in human strains; sulfamethoxazole/trimethoprim (SXT): 60.9% resistance in animal isolates, compared to 45.7% in the human population; ceftazidime (CAZ): 55.6% resistance in animals, compared to 43.3% in humans; cefotaxime (CTX): 54.9% of strains of animal origin were resistant, while only 25.2% of human strains showed the same profile; cefepime (FEP): 50.9% resistance in animal strains, compared to 9.5% in humans. The only exception to this trend was represented by amikacin (AMK), to which all *E. coli* strains isolated from animal samples were susceptible (0% resistance).

These results support the hypothesis that *E. coli* strains from animal sources generally exhibit a higher degree of resistance to multiple classes of antibiotics, which can be explained by the constant exposure to antimicrobial treatments in the veterinary and agricultural sectors, often used for prophylactic or performance-enhancing purposes. It is also worth mentioning that the prevalence of antibiotic resistance of *E. coli* strains isolated from human patients was below the 50.0% threshold for all antimicrobial substances tested. This observation indicates a relatively favorable susceptibility profile compared to strains from animal populations, where resistance rates above 50% were frequently recorded.

Based on these findings, it can be stated that monitoring and control of antibiotic use in veterinary medicine plays a crucial role in preventing the transmission of resistance to humans, especially in the context of zoonoses or the genetic transfer of resistance elements through the food chain and the environment.

It is also worth highlighting that, though the total number of *E. coli* animal isolates was significantly lower compared to the strains isolated from human patients, the percentage of MDR strains was considerably higher in those of animal origin. This aspect is even more relevant as all strains from the animal population were isolated from ceca harvested from clinically healthy animals (Figure 6).

Whereas, *E. coli* strains isolated from human population were exclusively harvested from active pathological processes, from patients admitted to various hospital wards. For these isolates, AST was performed for therapeutic purposes, to guide anti-infective treatment.

Regarding the strains fully susceptible to all eight antimicrobial substances, it was found that only 15 strains (10.0%) of those from animals fell into this category, while none of the strains of human origin showed complete susceptibility to all classes of antimicrobials. This finding raises questions about the antimicrobial selection pressure in the veterinary sector, even among healthy animals, suggesting the need for stricter antibiotic use policies in the livestock sector.

## 3. Discussion

This study provides one of the first integrated One Health assessments of multidrug-resistant (MDR) *Escherichia coli* in north-western Romania, directly comparing isolates from hospitalized patients and food-producing animals. The consistently higher resistance rates in animal isolates—particularly from pigs—highlight the veterinary sector as a key reservoir of resistance determinants that may circulate across the human–animal interface. Our data show that 79.5% of animal isolates were MDR, compared with 70% of human isolates. Resistance to β-lactams (ampicillin, cefotaxime, cefepime) and fluoroquinolones (ciprofloxacin) was markedly greater in animal isolates, while amikacin retained full efficacy. More precisely, the isolates from departments such as Nephrology, Urology Internal Medicine and Neurology the showed the highest resistance to ampicillin, cefotaxime, ceftazidime, cefepime and ciprofloxacin, but yet lower in comparison with animal isolates. For human isolates, such patterns could highlight the role of patient population characteristics (e.g., chronic conditions, repeated hospitalizations) in driving higher multidrug resistance burdens in specific wards.

These results suggest that antimicrobial selection pressure remains stronger in livestock, even in clinically healthy animals, than in hospitalized human populations. The comparatively lower resistance rates in human isolates may reflect improved antimicrobial stewardship in hospital settings, whereas veterinary use remains less regulated, with continued prophylactic and metaphylactic applications.

The results obtained in this study are consistent with findings reported by other researchers, showing similar trends of higher antimicrobial resistance in *Escherichia coli* strains isolated from animal populations compared with those from humans. For example, Nhung N.T. et al. investigated phenotypic resistance and potential interspecies transmission of *E. coli* between broiler chickens and humans in households from the Mekong Delta region of Vietnam. Their analysis, based on 237 chicken farms and 426 residents, yielded 385 *E. coli* strains from humans and 237 from chickens. They reported a higher prevalence of MDR strains in chicken isolates (63.3%) compared with human isolates (55.1%), a trend also observed in our study. Notably, cephalosporin resistance was significantly more prevalent in avian strains than in human strains, across all compounds tested [32]. These findings reinforce the hypothesis that the use of antimicrobials in livestock, including in clinically healthy animals, contributes to the selection and maintenance of resistant bacterial strains with zoonotic potential. The Vietnamese study therefore validates our observations of increased AMR levels in animal isolates and highlights the need for stricter control measures in antibiotic use on farms, along with systematic monitoring of resistant bacterial circulation between humans and animals. Comparable results were also reported by Kiet B.T. (2022) [33], who evaluated the prevalence of AMR in *E. coli* strains from chickens and humans, with isolates obtained from healthy subjects. Their study focused on five antimicrobials commonly used in poultry farming (ciprofloxacin, enrofloxacin, doxycycline, gentamicin, and florfenicol). The prevalence of resistance was consistently higher in chicken isolates compared to human isolates. For ciprofloxacin, resistance rates were 3.88% in chickens versus 0.054% in humans; for enrofloxacin, 6.99% versus 0.08%; for doxycycline, 20.60% versus 4.46%; for gentamicin, 10.03% versus 0.21%; and for chloramphenicol, 41.48% versus 7.65 [33]. These data are comparable to those obtained in our study for most antimicrobial substances tested, further confirming the broader pattern of elevated resistance among animal isolates.

Additional support for these observations comes from a study conducted by Kamonwan et al. in northern Thailand, which examined *E. coli* isolates from healthy pigs raised on farms of varying sizes and from humans in direct contact with these animals. The prevalence of antimicrobial resistance averaged 26.0% in pig isolates compared with only 13.0% in human isolates. Interestingly, the level of resistance among humans was not significantly affected by farm size or the degree of direct animal contact [34]. Taken together, these studies corroborate our findings and reinforce the notion that *E. coli* strains isolated from animal populations—often from clinically healthy individuals—tend to exhibit higher levels of antimicrobial resistance compared with human-derived isolates, which are frequently collected from clinical cases. This underscores the need for continuous AMR surveillance along the animal–human–environment interface and the implementation of strategies to ensure prudent antimicrobial use, particularly within the veterinary sector.

When comparing resistance profiles across the three interconnected sectors (human, animal, and environmental) clear differences and links emerge within the Romanian context. In our study, the prevalence of *E. coli* in broiler sources was 100% (27 isolates from 27 samples). The high isolation rate of *E. coli* in broilers significantly contrasts with the 30% overall prevalence reported in a previous study from Romania concerning *E. coli* in retail chicken meat [35]. On the other hand, the results of Antimicrobial Susceptibility Testing for broilers agreed with the overall findings of that study, except for gentamicin. In particular, 0% of the broiler isolates exhibited gentamicin resistance, which is less than the 40% resistance observed in the study’s retail chicken meat. The early colonization of *E. coli* in the digestive tract of poultry, even from hatchery, may be the cause of the high existence of the bacterium in chickens [36]. Different isolation methods, sample selection (number and type of samples from healthy and/or diseased birds), geographic variations, management of chicken farms, and different antibiotic dosing ranges could all be responsible for the variations in prevalence. Based on the available information, in Romania, at that current period of time from 2022–2023, there is little information available regarding the prevalence of *E. coli* isolated from poultry samples.

Another study conducted by Maciuca et al. in Romania also showed that the poultry production chain played an important role as a reservoir for resistant *E. coli*. In that study, 69% of broilers carried ESBL/*Amp*C-producing isolates, and more than half of them harbored the *blaCTX-M-15* gene, which was also predominant among human clinical isolates. Although no direct clonal overlap between human and poultry strains was reported by the authors, they stressed the possibility of resistance dissemination through plasmid transfer at the food-chain level [37]. Such findings are closely aligned with the higher values observed in the present study for food-producing animals-especially pigs compared with those from hospitalized patients. These data together reinforce the One Health concern that the circulation of resistance determinants between animals and humans via shared ecological and food-chain pathways may take place, even when direct clonal transmission is not detectable. Therefore, coordinated monitoring and stricter antimicrobial stewardship within the livestock sector remain key measures for reducing the risk of resistant *E. coli* entering human populations.

A comparison of antimicrobial resistance in the current study from the 2022–2023 period with more recent data on *E. coli* isolates from poultry in north-western Romania indicates a clear progressive intensification of resistance patterns. Our results for the 2022–2023 period showed that *E. coli* strains isolated from animals already exhibited a high prevalence of multidrug resistance (MDR = 79.7%), with resistance rates higher than 50% also for key antimicrobials. In 2025, the study by Rus et al. showed further consolidation of these resistance traits among poultry isolates: in that survey, more than half of all *E. coli* strains were classified as XDR (XDR = 50%) and another 44% as MDR, while fully susceptible isolates accounted for only 6% [9]. The emergence of antibiotic-resistant bacterial strains and their transmission to humans pose a growing threat to food safety and public health, as shown by the steady increase in MDR and XDR *E. coli* from 2022 to 2025. Therefore, there is an urgent need to explore natural and nanomaterial-based products, as sustainable alternatives to antibiotics in poultry production [38].

In the current study, the prevalence rate of *E. coli* was 100% in pigs (93 isolates from 93 samples). In another study in Romania showed that across all antimicrobials with overlap, pig source *E. coli* demonstrated consistently high resistance to the β-lactams, fluoroquinolones, and SXT in both studies, while amikacin exhibited complete efficacy. Isolates of porcine origin showed markedly higher resistance, especially against β-lactams, fluoroquinolones, and inhibitors of the folate pathway, whereas aminoglycosides, most notably amikacin, showed full retained activity, and pigs thus appear to be a primary reservoir for MDR strains [39].

Within this research, resistance among *E. coli* isolated from food-producing animals has been consistently higher compared to those isolated from hospitalized human patients, and the highest burden was recorded in pigs, more precisely high resistance rates to ampicillin (82.8%), ciprofloxacin (65.6%), cefotaxime (58.1%), and sulfamethoxazole/trimethoprim (68.8%) in pigs by far exceeded the resistance level in human isolates, which is below 50% in all cases. These data emphasize that livestock production systems are strong selective environments for the emergence of antimicrobial resistance. When comparing our results with those from the previous studies in Western Romania (2020) [27], it can be observed a similar trend: pig isolates proved to be significantly more resistant than human isolates, for most classes of antibiotics tested, especially β-lactams and fluoroquinolones. Although resistance to ciprofloxacin in pigs was reported in both studies, our study revealed a marked increase compared to the previous one from ~28% in 2020 to 65.6% in 2022–2023, reflecting a negative resistance trend in swine farms and implying continued selective pressure due to veterinary use of antibiotics. In humans, there was a slight increase in amikacin resistance, 2.85%, which may reflect differences in the use of aminoglycosides in the clinical and agricultural fields.

From an environmental perspective, Szekeres et al. (2017) [40] demonstrated that Romanian hospital wastewater contains high concentrations of antibiotic residues and resistance genes, including sulI and qacE delta 1, which persist even after treatment. These findings highlight wastewater effluents as important reservoirs and transmission pathways for resistance genes originating from clinical and veterinary sources [40].

Taken all three sectors together, the Romanian data show a continuum of antimicrobial resistance across sectors: human clinical isolates reflecting hospital antibiotic use patterns; animal isolates showing intensive exposure to veterinary antimicrobials; and environmental reservoirs sustaining and disseminating resistance determinants. This integrated perspective reinforces the epidemiological significance of cross-sectoral AMR circulation and underscores the need for coordinated One Health surveillance, improved waste management, and stricter antimicrobial stewardship policies in both healthcare and agriculture [41].

The hypothesis that *E. coli* strains producing extended-spectrum β-lactamases (ESBLs) or *amp*C enzymes in animals may cause infections in humans has been frequently raised, though not definitively confirmed [41,42,43,44]. Nevertheless, numerous molecular epidemiology studies have identified significant genetic similarities between *E. coli* strains isolated from animals, animal-derived products, and humans, suggesting possible epidemiological links [45,46,47]. Antibiotic resistance is increasing across species in parallel due to the simultaneous use of antibiotics in veterinary and human treatment. The persistence and spread of resistance are further supported by horizontal gene transfer, since resistance genes can spread both directly through interspecies contact or the ingestion of tainted animal products and indirectly through environmental contamination from animal excretions containing antimicrobial residues or unregulated farm use of antibiotics. The dissemination of mobile genetic elements, such as plasmids and transposons, that carry AMR determinants, as well as the selection of resistant strains, are facilitated by subsequent exposure of humans and other species to such conditions [48,49].

The consistently higher levels of resistance found in livestock isolates as opposed to those from human populations may be explained by a number of factors. Antimicrobials are commonly used in food-producing animals for prophylactic, therapeutic, and, in certain situations, growth-promoting purpose. Even when used in clinically healthy herds and flocks, these practices create selective pressure that favors the emergence and persistence of resistant strains [42]. Moreover, intensive farming conditions, characterized by high stocking densities and frequent antimicrobial exposure, provide an environment conducive to the amplification and horizontal transfer of resistance genes [43]. Another important factor is environmental contamination: resistant bacteria and antibiotic residues from animal waste can find their way into soil and water systems, forming reservoirs of resistance that could eventually impact human populations through direct contact, food, or water [48].

The overlap in resistance phenotypes between animal and human isolates reinforces the interconnectedness of AMR emergence across sectors. Animal waste, contaminated food products, and environmental runoff can act as conduits for resistant bacteria or genes to reach human populations. Within a One Health framework, these findings emphasize that effective AMR control requires coordinated policies: (i) restriction of non-therapeutic antibiotic use in livestock, (ii) harmonized AMR surveillance integrating human, animal, and environmental data, and (iii) awareness programs for both veterinarians and physicians regarding responsible antimicrobial prescribing.

These findings have ramifications that go beyond veterinary care and emphasize how urgently a One Health approach is needed. Meat and other food products derived from animals may become contaminated by resistant *E. coli* strains that are circulating in livestock, providing a direct route of transmission to humans [49]. In addition, resistance determinants can move across species boundaries through mobile genetic elements, potentially disseminating to pathogenic bacteria of clinical relevance in humans [44]. Integrated surveillance systems that track trends in antibiotic use and resistance in the human, animal, and environmental domains are necessary to address this problem. It is crucial to implement policies that encourage the prudent use of antibiotics, such as prohibitions on non-therapeutic uses in livestock and the creation of antibiotic substitutes in agriculture [43]. Public health strategies should also include educational campaigns for both healthcare professionals and farmers, emphasizing the risks of inappropriate antibiotic use [41].

This study has some limitations that need to be considered. First, it focused only on one region (Satu Mare County, Romania), which may limit how well the findings apply to other areas. Second, despite our thorough examination of phenotypic resistance patterns, we did not conduct molecular analysis of resistance genes or clonal lineages, which would have offered a more profound understanding of resistance mechanisms and potential pathways of transmission. Despite these drawbacks, our research offers valuable comparative information on MDR *E. coli* from animals and humans in the same area. Molecular techniques for identifying resistance genes and plasmid types should be incorporated into future research. Additionally, incorporating environmental samples like soil and water would contribute to the development of a more comprehensive picture of antimicrobial resistance throughout the animal–human–environment interface.

Our results demonstrate that food-producing animals, especially pigs, are significant reservoirs of MDR *E. coli* in north-western Romania. The alignment of phenotypic resistance patterns between human and animal isolates indicates potential exchange of resistant strains or genes across sectors. Strengthened surveillance, stricter regulation of veterinary antibiotic use, and integration of molecular epidemiology are crucial next steps toward mitigating AMR dissemination at the human–animal–environment interface.

## 4. Materials and Methods

### 4.1. Study Design and Setting

We conducted a cross-sectional, One Health comparative study in Satu Mare County, Romania (January 2022–December 2023). In total, 701 samples were collected: 550 clinical specimens from hospitalized patients and 151 cecal samples from food-producing animals (turkeys, broiler chickens, pigs). Out of these, 571 non-duplicate *Escherichia coli* isolates were recovered (420 human; 151 animal) and subjected to antimicrobial susceptibility testing (AST).

### 4.2. Human Sampling and Isolate Collection

Clinical specimens were obtained as part of routine care from 21 departments of Satu Mare County Emergency Hospital (e.g., Cardiology, Dermatology, Gastroenterology, Gynecology, ICU, Infectious Diseases, Internal Medicine, Nephrology, Neurology, Oncology, Orthopedics, Pediatrics, Rheumatology, Surgery, Urology). Specimen types included feces, urine, blood, sputum, ear secretions, cerebrospinal fluid, purulent wound secretions, and puncture fluids. Department-level counts are provided in Table 5. Specimen-level data were not available for each department; counts are presented at the department level as provided by the hospital laboratory. A total of 420 *E. coli* isolates were recovered from 550 human samples. Inclusion was based on clinical suspicion of each attending clinicians; only one isolate per patient per episode was retained.

### 4.3. Animal Sampling and Isolate Collection

Ceca were collected at local slaughterhouses serving farms in north-west Romania from turkeys (*n* = 31), broilers (*n* = 27), and pigs (*n* = 93). Animals were raised for commercial meat production; licensed veterinarians performed ante-mortem inspections and found no clinical signs of disease. All samples were obtained from animals raised on conventional farms, located in 16 counties across Romania. Information on prophylactic or therapeutic antibiotic use at the farm level was not available, as these data were not systematically recorded or reported. Ceca were harvested immediately post-slaughter, kept refrigerated, transported to the laboratory the same day, and processed within 24 h. In total, 151 *E. coli* isolates were obtained from animal samples.

### 4.4. Isolation and Identification of E. coli

Samples requiring enrichment (e.g., feces, biological fluids, other secretions) were incubated in Buffered Peptone Water (Thermo Scientific™, Dreieich, Germany) at 37 °C for 18 h, then streaked onto MacConkey agar (Thermo Scientific™) and incubated at 37 °C for 18–24 h. Samples not requiring enrichment (e.g., purulent secretions, sputum) were plated directly on MacConkey agar. Presumptive *E. coli* colonies were sub-cultured at least twice on nutrient agar to ensure purity and then confirmed by standard biochemical testing: Triple Sugar Iron (TSI) agar and Simmons citrate agar (Thermo Scientific™, Dreieich, Germany), followed by API 20E identification system (bioMérieux, Lyon, France). One colony per sample was processed to ensure pure cultures.

### 4.5. Antimicrobial Susceptibility Testing (AST)

Human isolates were tested by standardized Kirby–Bauer disc diffusion on Mueller–Hinton agar (Thermo Scientific™, Dreieich, Germany). Bacterial suspensions were adjusted to 0.5 McFarland and plates incubated at 37 °C for 18–24 h; inhibition zones were interpreted according to CLSI M100, ed. 35 (2025) and/or EUCAST v15.0 (year 2025) breakpoints, and the strains were classified as resistant (R) or susceptible (S) [30,31].

Animal isolates were tested by broth microdilution using Sensititre™ panels with automated inoculation (Thermo Scientific™, Dreieich, Germany). From 24 h cultures, 3–4 colonies were suspended to 0.5 McFarland in sterile water; 10 µL were transferred to cation-adjusted Mueller–Hinton broth buffered with TES. Plates (50 µL antimicrobial + 50 µL inoculum per well) were sealed and incubated at 35 °C for 20 h. Minimum inhibitory concentrations (MICs) were read using a Biomic V3 analyzer (Giles Scientific Inc., San Diego, CA, USA) and interpreted using the same breakpoint standard(s) as above.

All isolates (human and animal) were tested against: amikacin (AMK), ampicillin (AMP), cefotaxime (CTX), ceftazidime (CAZ), cefepime (FEP), ciprofloxacin (CIP), gentamicin (GEN), and sulfamethoxazole/trimethoprim (SXT) (Bio-Rad, Hercules, CA, USA), The selection of the antimicrobial substances was based on the Decision (EU) 2020/1729 of 17 November 2020, which specifies the antimicrobial classes to be included in national and EU-level AMR surveillance programs, ensuring comparability of data across countries and sectors Quality control included *E. coli* ATCC 25922.

### 4.6. Resistance Category Definitions

Phenotypic categories followed Magiorakos et al. (2012) [50]:MDR (multidrug-resistant): non-susceptible to ≥1 agent in ≥3 antimicrobial categories.XDR (extensively drug-resistant): non-susceptible to all but ≤2 antimicrobial categories.Fully susceptible: 100% susceptible to all tested antimicrobial substances.

For clarity and comparability, non-standard term “partially resistant” was avoided and non-MDR isolates were reported as “non-MDR” [50].

### 4.7. Statistical Analysis

Associations between host category (human, turkey, broiler, pig) and resistance outcomes were assessed with χ^2^ tests (two-sided; α = 0.05). Where assumptions were borderline, we verified results with Fisher’s exact test. Data handling and figures were produced in Python 3.11.8 (Python Software Foundation, Wilmington, DE, USA) (packages: pandas, numpy, matplotlib; SciPy for statistics).

## 5. Conclusions

Classification of resistance showed that 70% of the *E. coli* isolates from humans were MDR, and no strain was 100% susceptible to all tested antimicrobials. Compared to human isolates, in strains isolated from animals, 79.47% of the strains were MDR, 11.25% were non-MDR, while 9.27% of strains were fully susceptible.

For AMP, CTX, CIP, GEN, SXT and FEP, statistically significant differences were recorded (*p* < 0.05), indicating the isolates from pigs showed significantly higher resistance rates to these antibiotics compared to those isolated from humans or birds. Thus, the pigs could represent an important reservoir of multidrug-resistant *E. coli* strains, which requires differentiated measures for surveillance and control of antimicrobial resistance, according to the One Health strategy.

Although *E. coli* strains isolated from animals came from clinically healthy individuals and strains from humans were isolated from pathological processes, the prevalence of antibiotic resistance was higher in the animal population than in human population, for all antimicrobials studied.

In conclusion, the link between antibiotic consumption and the emergence and development of antimicrobial resistance, both in animal and human populations, are bidirectional connections, as increasing antibiotic use in one population induces an increase in antibiotic resistance both in that population and indirectly in the other, all of which are potentiated by the effect that antibiotics have in the environment.

## Figures and Tables

**Figure 1 antibiotics-14-01157-f001:**
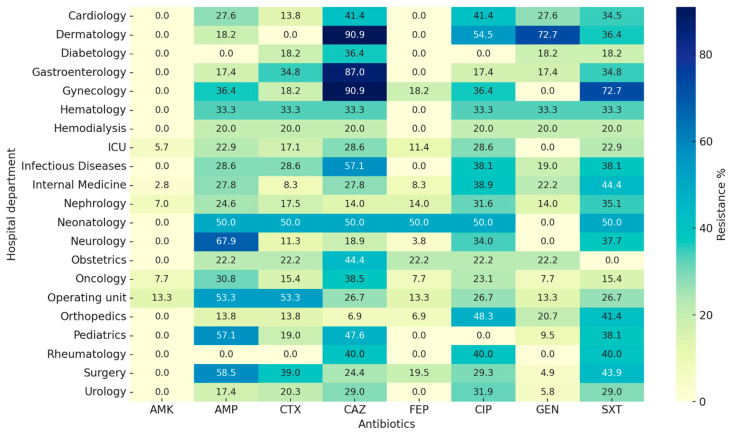
Heatmap showing the prevalence of antimicrobial resistance (%) in *E. coli* isolates across hospital departments. Legend: Darker shades correspond to higher resistance rates, enabling direct comparison of susceptibility profiles by ward and antibiotic; AMK (amikacin); AMP (ampicillin); CTX (cefotaxime); CAZ (ceftazidime); FEP (cefepime); CIP (ciprofloxacin); GEN (gentamicin); SXT (sulfamethoxazole/trimethoprim).

**Figure 2 antibiotics-14-01157-f002:**
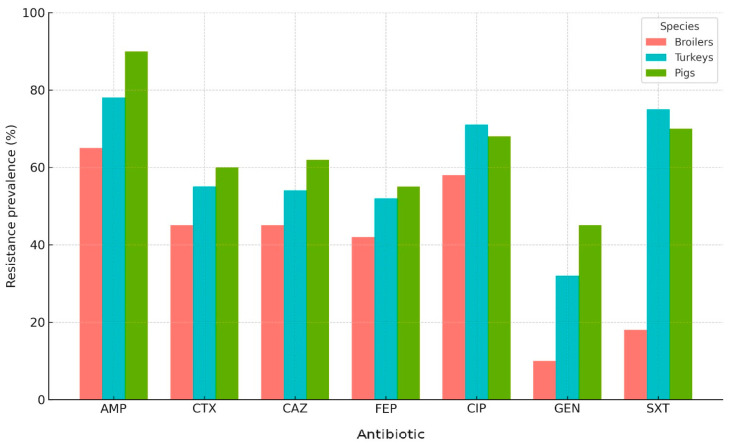
Stacked bar chart showing the prevalence of antimicrobial resistance in *E. coli* isolates from broilers, turkeys, and pigs. Legend: AMP (ampicillin); CTX (cefotaxime); CAZ (ceftazidime); FEP (cefepime); CIP (ciprofloxacin); GEN (gentamicin); SXT (sulfamethoxazole/trimethoprim).

**Figure 3 antibiotics-14-01157-f003:**
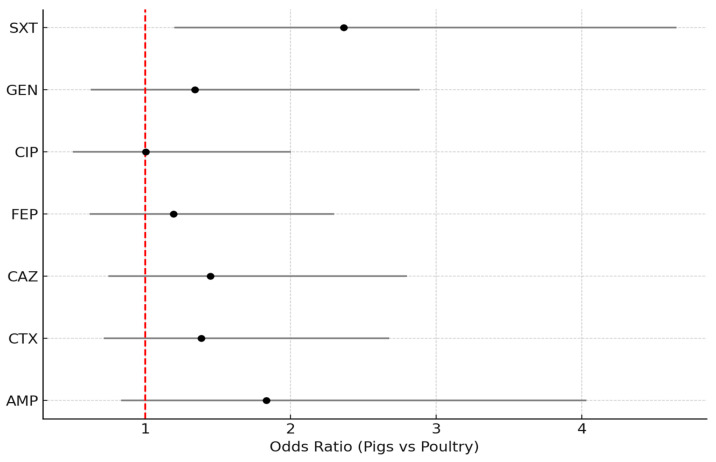
Forest plot of odds ratios (OR) comparing the likelihood of resistance in pig isolates versus poultry (broilers + turkeys) across antibiotics. Legend: points represent OR estimates; horizontal bars show 95% confidence intervals. The vertical dashed line indicates OR = 1 (no difference); AMP (ampicillin); CTX (cefotaxime); CAZ (ceftazidime); FEP (cefepime); CIP (ciprofloxacin); GEN (gentamicin); SXT (sulfamethoxazole/trimethoprim).

**Figure 4 antibiotics-14-01157-f004:**
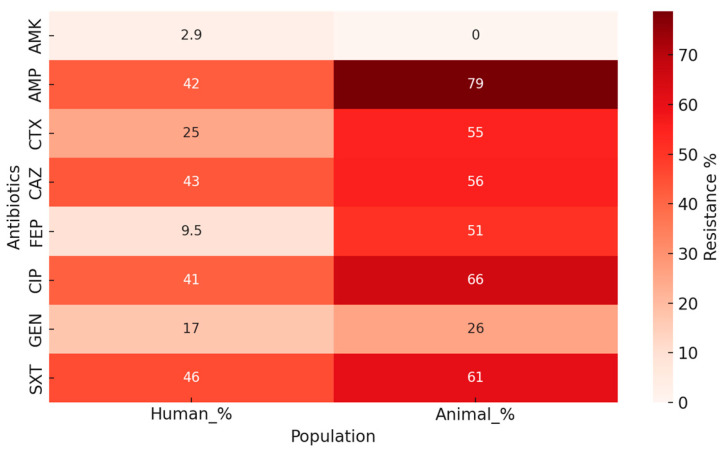
Heatmap of AMR prevalence (%) in *E. coli* isolates from humans and animals. Legend: Darker shades indicate higher levels of resistance, allowing visual comparison of susceptibility profiles between populations. AMK (amikacin), AMP (ampicillin), CTX (cefotaxime), CAZ (ceftazidime). FEP (cefepime), CIP (ciprofloxacin), GEN (gentamicin), SXT (sulfamethoxazole/trimethoprim).

**Figure 5 antibiotics-14-01157-f005:**
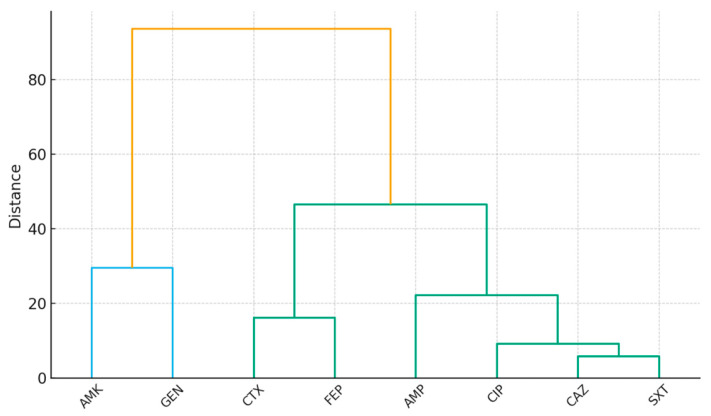
Dendrogram of antibiotic resistance clustering based on resistance prevalence in human and animal *E. coli* isolates. Legend: Horizontal distances reflect similarity in resistance patterns, with shorter distances indicating antibiotics that display comparable resistance rates across both populations; AMK (amikacin); AMP (ampicillin); CTX (cefotaxime); CAZ (ceftazidime); FEP (cefepime); CIP (ciprofloxacin); GEN (gentamicin); SXT (sulfamethoxazole/trimethoprim).

**Figure 6 antibiotics-14-01157-f006:**
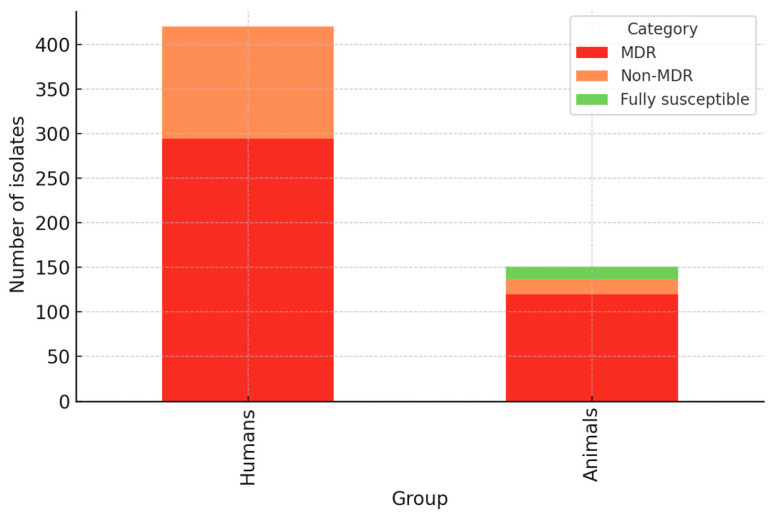
Stacked bar chart showing the distribution of multidrug-resistant (MDR), non-MDR and fully susceptible *E. coli* isolates in humans and animals. Legend: Bars represent the absolute number of isolates per group, divided into MDR (red), non-MDR (orange), and fully susceptible strains (green).

**Table 1 antibiotics-14-01157-t001:** Origin of isolated *E. coli* strains.

Department/ Category	Samples (No.)	Isolated Strains (No.)
Humans		
Cardiology	29	26
Dermatology	11	10
Diabetology	11	6
Gastroenterology	23	20
Gynecology	11	10
Hematology	3	2
Hemodialysis	5	2
ICU	35	22
Infectious Diseases	21	14
Internal Medicine	72	64
Nephrology	57	48
Neonatology	4	2
Neurology	53	46
Obstetrics	9	4
Oncology	26	16
Operating unit	15	8
Orthopedics	29	20
Pediatrics	21	20
Rheumatology	5	2
Surgery	41	34
Urology	69	44
Total	550	420
**Animals**		
Broilers	27	27
Turkeys	31	31
Pigs	93	93
Total	151	151
**Total humans + animals**	**701**	**571**

**Table 2 antibiotics-14-01157-t002:** Origin of *E. coli* isolates from patients according to age and gender.

Age	Gender		Total
	F	M	No. (%)
Newborns	2	0	2 (0.47)
1–12 months	10	6	16 (3.8)
1–17 years	6	0	6 (1.42)
18–40 years	8	8	16 (3.8)
41–65 years	62	46	108 (25.71)
>65 years	180	92	272 (64.76)
**Total**	**268**	**152**	**420 (100)**

**Table 3 antibiotics-14-01157-t003:** Resistance profile of *E. coli* strains isolated from patients.

Department	AMK	AMP	CTX	CAZ	FEP	CIP	GEN	SXT
Cardiology	0	8	4	12	0	12	8	10
Dermatology	0	2	0	10	0	6	8	4
Diabetology	0	0	2	4	0	0	2	2
Gastroenterology	0	4	8	20	0	4	4	8
Gynecology	0	4	2	10	2	4	0	8
Hematology	0	1	1	1	0	1	1	1
Hemodialysis	0	1	1	1	0	1	1	1
ICU	2	8	6	10	4	10	0	8
Infectious Diseases	0	6	6	12	0	8	4	8
Internal Medicine	2	20	6	20	6	28	16	32
Nephrology	4	14	10	8	8	18	8	20
Neonatology	0	2	2	2	2	2	0	2
Neurology	0	36	6	10	2	18	0	20
Obstetrics	0	2	2	4	2	2	2	0
Oncology	2	8	4	10	2	6	2	4
Operating unit	2	8	8	4	2	4	2	4
Orthopedics	0	4	4	2	2	14	6	12
Pediatrics	0	12	4	10	0	0	2	8
Rheumatology	0	0	0	2	0	2	0	2
Surgery	0	24	16	10	8	12	2	18
Urology	0	12	14	20	0	22	4	20
**TOTAL**	**12**	**176**	**106**	**182**	**40**	**174**	**72**	**192**
**AMR prevalence (%)**	**2.85**	**41.90**	**25.23**	**43.33**	**9.52**	**41.42**	**17.14**	**45.71**

Legend: AMK (amikacin), AMP (ampicillin), CTX (cefotaxime), CAZ (ceftazidime). FEP (cefepime), CIP (ciprofloxacin), GEN (gentamicin), SXT (sulfamethoxazole/trimethoprim), AMR (antimicrobial resistance). Numbers indicate the count of *E. coli* isolates resistant to each antibiotic per hospital department. Overall AMR prevalence (%) for each antibiotic is shown in the last row. Antimicrobial susceptibility was interpreted according to CLSI M100, ed. 35 (2025) and/or EUCAST v15.0 guidelines [30,31].

**Table 4 antibiotics-14-01157-t004:** Resistance profile of *E. coli* strains isolated from animals.

Species	AMK	AMP	CTX	CAZ	FEP	CIP	GEN	SXT
Broilers	0	17	12	12	11	17	0	5
Turkeys	0	25	17	17	17	21	13	23
Pigs	0	77	54	55	49	61	26	64
**TOTAL**	**0**	**119**	**83**	**84**	**77**	**99**	**39**	**92**
**AMR prevalence (%)**	**0**	**78.80**	**54.96**	**55.62**	**50.99**	**65.56**	**25.82**	**60.92**

Legend: AMK (amikacin), AMP (ampicillin), CTX (cefotaxime), CAZ (ceftazidime). FEP (cefepime), CIP (ciprofloxacin), GEN (gentamicin), SXT (sulfamethoxazole/trimethoprim), AMR (antimicrobial resistance). Numbers indicate the count of *E. coli* isolates resistant to each antibiotic per hospital department. Overall AMR prevalence (%) for each antibiotic is shown in the last row. Antimicrobial susceptibility was interpreted according to CLSI M100, ed. 35 (2025) and/or EUCAST v15.0 guidelines [30,31].

**Table 5 antibiotics-14-01157-t005:** Origin and number of human samples for the identification of *E. coli* strains.

Department	Samples (No.)	% of Total (*N* = 550)
Cardiology	29	5.3
Dermatology	11	2
Diabetology	11	2
Gastroenterology	23	4.1
Gynecology	11	2
Hematology	3	0.5
Hemodialysis	5	0.9
ICU	35	6.4
Infectious Diseases	21	3.8
Internal Medicine	72	13.1
Neonatology	4	0.7
Nephrology	57	10.3
Neurology	53	9.6
Obstetrics	9	1.6
Oncology	26	4.7
Operating Unit	15	2.7
Orthopedics	29	5.2
Pediatrics	21	3.8
Rheumatology	5	0.9
Surgery	41	7.4
Urology	69	12.5
**Total**	**550**	**100**

## Data Availability

All data generated or analyzed during this study are included in the submitted version of the manuscript.

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
