# Peer review of "A One Health Comparative Study of MDR Escherichia coli Isolated from Clinical Patients and Farm Animals in Satu Mare, Romania"

_antibiotics, 2025, doi:10.3390/antibiotics14111157_

Round 1
Reviewer 1 Report
Comments and Suggestions for Authors
A One Health comparative study of MDR Escherichia coli 2 isolated from clinical patients and farm animals in Satu Mare, 3 Romania
The study aimed to compare antimicrobial resistance (AMR) profiles of E. coli isolates from hospitalized patients and food-producing animals in Satu Mare, a county located in north- western Romania. The study underscores the veterinary sector as a key contributor to the maintenance and spread of MDR E. coli, and the need for stricter antimicrobial use policies in livestock to reduce transmission risks across human and animal populations.
Abstract:
- Could you please specify the source of the coli isolates recovered in hospitals (e.g., swab samples, stool, instruments, etc.)? I would also appreciate the same information regarding the animal isolates.
- Regarding statistical analysis, significant and non-significant differences (P < 0.05) should be interpreted and reported in the abstract when comparing outcome results.
Introduction:
- Could you please clarify the definition of the term 'One Health challenge' and provide the source of this terminology?
- Objectives: Could you please clarify why the investigators chose to study only one bacterial strain ( coli)? What was the rationale for excluding other Gram-negative bacteria such as Salmonella or Campylobacter, as well as Gram-positive bacteria like Staphylococcus or Clostridium? It would have been preferable for the investigators to include at least one additional Gram-negative and/or Gram-positive bacterial strain to strengthen and support their findings.
- Line 89: In addition to reducing antibiotic use, there are other strategies that can help control the transfer of resistance genes. Could you please specify which approaches you consider essential for limiting the spread of antimicrobial resistance, and explain how these align with your core objectives and findings?
- Line 95: what about cattle farms?
- Line 101: Therefore, this study “could represent” one of the first systematic…
- Objectives: A major concern is that the investigators studied antimicrobial resistance patterns only phenotypically. It is essential to perform genetic identification of resistance mechanisms and correlate those findings with resistance patterns in both human and animal isolates. It was essential to identify the antimicrobial resistance genes and correlate them with the phenotypic results for both human and animal isolates.
Methodology:
- Table 5: Could you please specify the number and percentage of clinical samples according to specimen type?
- Line 419 mentions 550 human samples, whereas Table 5 indicates 571 human samples. Could you please clarify this discrepancy.
- Could you please specify the source of the coli isolates (specimen type) recovered from animals?
- Could you please clarify why the number of animal samples differed significantly?
- Could you please provide the reference for the isolation and identification of coli? Additionally, was it important to include advanced verification procedures (PCR) to identify E. coli and establish genetic correlations between different strains?
- Antimicrobial susceptibility testing: Could you please clarify why the authors used minimum inhibitory concentrations (MICs) for animal isolates? Did they assess antimicrobial resistance using CLSI breakpoints for animal isolates as they did for human isolates?
- Line 453: Could you please clarify the reason for choosing these antibiotics to assess antimicrobial resistance?
- What was the source of the antibiotic discs?
- What was the concentration of each antibiotic used?
- Resistance category definitions: Could you please provide the definition of MDR according to CLSI (2025)?
- How many independent experimental trials were conducted to perform this study?
Results and discussion
- Line 112: Please note that the 571 samples reported in Table 5 were collected from human specimens and are not related to the animal samples mentioned in lines 112-113. Table 5 specifically reports samples collected from human specimens, totaling 571 samples. Is that correct? Please clarify and re-check the sample numbers throughout the manuscript.
- Table 3: Please clarify what the numbers presented represent, as it is not clear. Also, please provide the reference for the resistance profile? I would also appreciate the same regarding Table 4.
- The Results section should focus solely on presenting the major findings; however, the authors have included extensive discussion in this section, which should be moved to the Discussion section!
- The authors should clearly explain why animal isolates consistently display higher resistance levels than human isolates and correlate their findings with existing published data?
- A more thorough discussion and interpretation of the results should be provided in the Discussion section, with appropriate use of statistical analysis and data interpretation! The current discussion reads more like a literature review than a focused interpretation of the study's findings!
- As mentioned earlier, a major concern is that the investigators did not perform gene analysis to study antimicrobial resistance genes, correlate them with phenotypic results, or explore genetic relationships between different human and animal isolates.
Conclusion:
- It is clear that 30% of human isolates were non-MDR, given that 70% were MDR. Please consider deleting the phrase ‘30% were non-MDR.’
- Since 79.74% of the animal strains were MDR, approximately 20% were considered non-MDR. Could you please clarify the status of the 9.68% identified as non-MDR isolates?
- In general, the conclusion should be concise and directly related to the study findings. Could you please rewrite the conclusion to be brief, precise, and free of repetition?
Author Response
Please see attachement.

Reviewer 2 Report
Comments and Suggestions for Authors
The manuscript addresses the important and timely issue of multidrug resistance (MDR) in Escherichia coli strains isolated from both human clinical cases and farm animals. The study is relevant within the context of the One Health approach and provides valuable data for both human and veterinary medicine.
Overall, the article is well structured, and the data presented are of scientific interest. However, several critical issues should be addressed to improve the clarity, completeness, and scientific rigor of the manuscript:
- Abstract (Line 27). Phrases such as "in animal isolates…” and "In human samples…” should be revised to clearly state that resistance was observed in coli strains isolated from animals and humans, respectively - not in the samples themselves.
- It is generally not recommended to use abbreviations (e.g., MDR) in keywords. The authors are advised to spell out the terms and consider including specific antibiotics to which resistance was observed.
- Please explain all abbreviations that may not be universally known. Uncommon acronyms should be defined at first use.
- Antimicrobial Use in Romania. The Introduction would be strengthened by including background information on commonly used antimicrobials in both human and veterinary medicine in Romania. Furthermore, a discussion on possible cross-usage (shared compounds or classes) should be included in the Discussion section to support the One Health perspective.
- Ethical Approval. As the study involves both human subjects and animals, it is mandatory to include detailed information regarding ethical approval. Please provide the name of the ethics committee, approval number, and confirm whether informed consent was obtained from human participants. Without this information, the manuscript is not suitable for publication.
- Section 4.2 – Human Sampling. Please clarify the criteria for inclusion and exclusion of patients. This is essential to assess the reliability and reproducibility of the study.
- Line 436 – Identification of coli. Clarify the method of species identification. Was molecular or biochemical confirmation performed? How was strain purity ensured? If only culture-based methods were used, the claim of E. coli identification may be questionable. If the strains were deposited in a microbial culture collection, please provide accession or collection numbers.
- Lines 453–455 – Antibiotic Panel. Please provide a rationale for the selection of antibiotics used in susceptibility testing. This should be discussed in the Discussion section and supported by references.
- Results – Tables 1 and 2. Consider merging Tables 1 and 2 into a single, more detailed table for better readability and comparison between human and animal isolates.
- Clinical Department Information. The rationale for listing the medical departments (e.g., Hematology, Gastroenterology, Dermatology) is unclear. If this is intended to link resistance patterns with treatment histories, additional data on administered antibiotics per department would be highly valuable. If such data are unavailable, consider omitting this classification or clarifying its relevance.
- Discussion of Table 3 (Section 2.2). Use the data from Table 3 as a basis for in-depth discussion. Is there a correlation between resistance profiles and the most frequently used antibiotics in specific clinical or veterinary settings?
- Lines 139–148. This content would be better placed in the Discussion section, where it could be expanded with references and integrated into a broader analysis of your findings.
- Discussion Section. Strengthen the Discussion by comparing your results with previously published data. Include a critical assessment of similarities, differences, and possible explanations for observed resistance trends.
- Conclusion Section. Revise the Conclusion to be more concise and focused. Summarize the key findings and implications in one clear paragraph. Currently, some content belongs more in the Discussion section.
- IRB and Informed Consent Statements. The manuscript currently states that the "Institutional Review Board Statement" and "Informed Consent Statement" are Not applicable. This is incorrect, as the study clearly involves human and animal subjects. This must be corrected to reflect proper ethical oversight.
Reviewer 3 Report
Comments and Suggestions for Authors
Below are my comments on the submitted manuscript
- The introduction is well defined; however, given the authors' intent to characterize the antimicrobial resistance (AMR) landscape in Romania, it would add significant value if the manuscript provided more detailed information on antimicrobial usage patterns in both animal and human sectors within Romania.
- In the methods section, please elaborate on whether the samples were obtained from conventional or non-conventional farms. Additionally, specify what classes and agents of antibiotics were administered as prophylactic treatments in livestock or poultry, as this can impact AMR profile.
- Clarify and provide the exact concentrations of antibiotics used for antimicrobial susceptibility testing (AST).
- The categorization for human samples should be based on specimen type rather than hospital department. If the categorization by department is retained, please include specimen type details for each department.
- In the results section, there is a discrepancy between Table 4 and Figure 2; for example, Table 4 reports ampicillin (AMP) resistance at 78%, while Figure 2 shows values exceeding 200, which may be misleading. Ensure all data are accurately harmonized and consistently presented across tables and figures.
- Figure 2 can be improved overall. Rather than a stacked bar chart, a simple bar chart with separate bars for each species would be more informative.
- It would be valuable to report the prevailing AMR profiles for each sector (human, animal, environmental), highlighting differences in resistance patterns and their epidemiological significance.
- The discussion section can be improved by integrating data on the specific types of antimicrobials used in both human and animal sectors in Romania and comparing associated resistance patterns. Consider discussing the implications for One Health strategies and stewardship interventions.
In line 77; influence is wrongly spelled
in line 399 E.coli is not italic.
Round 2
Reviewer 1 Report
Comments and Suggestions for Authors
I have reviewed the authors’ responses and the revised version of the manuscript. The authors have addressed the comments thoroughly, and I have no further comments at this time.
Reviewer 2 Report
Comments and Suggestions for Authors
The authors have thoroughly revised the manuscript, addressing all raised questions. I consider the manuscript suitable for publication in its current form.